# Distributive Justice as the Foundational Premise of Fair ML: Unification, Extension, and Interpretation of Group Fairness Metrics

## Abstract

Group fairness metrics are an established way of assessing the fairness of prediction-based decision-making systems. However, these metrics are still insufficiently linked to philosophical theories, and their moral meaning is often unclear. We propose a general framework for analyzing the fairness of decision systems based on theories of distributive justice, encompassing different established "patterns of justice" that correspond to different normative positions. We show that the most popular group fairness metrics can be interpreted as special cases of our approach. Thus, we provide a unifying and interpretative framework for group fairness metrics that reveals the normative choices associated with each of them and that allows understanding their moral substance. At the same time, we provide an extension of the space of possible fairness metrics beyond the ones currently discussed in the fair ML literature. Our framework also allows overcoming several limitations of group fairness metrics that have been criticized in the literature, most notably (1) that they are parity-based, i.e., that they demand some form of equality between groups, which may sometimes be harmful to marginalized groups, (2) that they only compare decisions across groups, but not the resulting consequences for these groups, and (3) that the full breadth of the distributive justice literature is not sufficiently represented.

## 1 Introduction

Supervised machine learning (ML) is increasingly being used for prediction-based decision making in various consequential applications, such as credit lending, school admission, and recruitment. Recent work has shown that the use of algorithms for decision making can reinforce existing biases or introduce new ones [8]. Consequently, fairness has emerged as an important desideratum for automated decision making. As recent cases in practice have shown, this is crucial in order to mitigate unjustified disadvantages towards certain demographic groups (see, e.g., [2, 46, 21, 40]). However, quantifying the fairness of decision making systems is not straightforward as any morally appropriate notion of fairness heavily depends on the given context.

Many different measures have emerged in the algorithmic fairness literature to assess and mitigate unfairness towards marginalized groups in decision making systems. Many of the proposed notions of fairness are in the category of so-called group fairness criteria [7], some of which are mathematically incompatible in practice Kleinberg et al. [32], Chouldechova [14]. Therefore, satisfying such a fairness criterion comes at the expense of not being able to satisfy others Kleinberg et al. [31], Wong [55]. ~~All~~Most existing group fairness criteria demand equality of a certain value between different socio-demographic groups [12]. However, our framework is also compatible with other notions of

fairness that concern groups of individuals, such as preference-based fairness [56, 30]. However, this stands, which is in contrast to the comparison of individuals, as it is done with other types of fairness such as individual fairness [18, 52], envy-freeness [6] or counterfactual fairness Kusner et al. [34]. Readers unfamiliar with group fairness may refer to [38, Chapter 2], [53], and [7] for an overview of the topic. We briefly introduce and formally define the most-discussed group fairness criteria in Appendix A.

Much of the algorithmic fairness literature evolves around a limited set of group fairness metrics and is often not clearly linked to the many philosophical theories of justice that have been well-discussed. Kuppler et al. [33] find that there is little to no overlap between philosophical theories of justice and metrics in the algorithmic fairness literature and conclude that "apparently, the fair machine learning literature has not taken full advantage of the rich and longstanding literature on distributive justice" [33, p. 17]. Therefore, the definitions of group fairness could be described as quite narrow when viewed from a philosophical perspective. This becomes evident when thinking about an example: Group fairness metrics typically demand that groups are equal with respect to *some* metric. Demanding equality between groups often makes sense, but consider a case in which we could increase the utility of one group without harming another: Should we do this? While we cannot say that this is always a good idea, it at least seems to be a reasonable objection to group fairness metrics, which demand equality at all costs. Therefore, this paper asks whether group fairness metrics can be extended to compare groups in other ways.

As of today, only a limited number of fairness metrics have been discussed, forcing stakeholders to choose between a set of pre-defined metrics that they then have to justify for their context. This paper, in contrast, presents a general framework for the derivation of targeted and context-specific fairness metrics, starting from values and moral views, and connects these to the philosophical literature, in particular to theories of distributive justice.

Our main contributions can be summarized as follows:

1. We propose a general framework for assessing the fairness of prediction-based decision systems, based on theories of distributive justice, and allowing for different established "patterns of justice" that correspond to different normative positions. The framework is based on an analysis of how utility is distributed between groups. "Pattern of justice" refers to normative ideas of what constitutes a just distribution.

2. We show that the most popular group fairness metrics can be interpreted as special cases of this approach, which thus establishes a unifying framework that includes established metrics, but also shows how new ones can be constructed.

We first present existing literature on group fairness (including its limitations) in Section 2. In Section 3, we present our unified framework for utility-based definitions of group fairness. We focus on the mathematical formalization of different aspects of the distributive justice literature while keeping the review of the philosophical side short. More details about the philosophical side can be found in the companion paper [3]. Section 4 then demonstrates that existing group fairness metrics are special cases of our utility-based approach. Finally, we discuss the implications of this and possible future work in Section 5.

## 2 Limitations of current group fairness criteria

Existing group fairness criteria pursue an egalitarian approach. This means that they demand equality of a certain value between different socio-demographic groups [12]. The fulfillment of these criteria is easy to assess, as this only requires access to a few variables (e.g., to check whether statistical parity is satisfied, we only need the decisions and the group membership of individuals). However, they also come with several limitations:

**The "leveling down objection"**     As has been shown by [27], in some cases, enforcing group fairness criteria can yield worse results for all groups in order to ensure parity between the groups. This is what is known as the "leveling down objection", which is often brought forward to challenge egalitarianism in philosophical literature [41, 17]: In a case in which equality requires us to worsen the outcomes for everyone, should we really demand equality or should we rather tolerate some

inequalities? As criticized by Cooper and Abrams [15], Weerts et al. [54], existing definitions of group fairness lack this differentiation as they always minimize inequality.

**No consideration of consequences**   As pointed out by Hertweck et al. [24] and Weerts et al. [54], a large part of the existing work on fairness criteria seems to focus on an equal distribution of favorable decisions and not on the consequences of these decisions. Binns [11] notes that these criteria "[assume] a uniform valuation of decision outcomes across different populations" [11, p. 6], and notes that this assumption does not always hold. Whether a loan approval has a positive effect on one's life or not arguably depends on one's ability to repay this loan (and possibly on other individual attributes). This narrow focus on the algorithm's decisions instead of its consequences makes it difficult to use existing group fairness criteria for a moral assessment of unfairness in decision making systems. Parity-based criteria that only consider the decisions but not their consequences do not allow us to deliberately give positive decisions to a larger share of the disadvantaged group as this would be a form of unequal treatment. However, Kasy and Abebe [29] argue that in such a case, unequal treatment can be required by justice to reduce overall inequalities. Several works have therefore taken a utility-based view of fairness. Heidari et al. [22]'s utility-based definitions of fairness focus on the effects of decisions while [13] developed a method that follows the Rawlsian leximin principle to increase the welfare of the worse off groups. However, none of them provides a general framework that encompasses different theories of distributive justice.

**Limited set of fairness definitions**   Another limitation of existing group fairness criteria is that they represent a limited set of alternatives. One has to choose one over the others, as they are mathematically incompatible [32, 14]. [47, 28] have highlighted that the criteria differ with respect to underlying moral values. Thus, solely choosing one among the limited set of criteria might fail adequately represent a morally appropriate definition of fairness for a given context. Heidari et al. [23] show how existing group fairness criteria can be viewed as instantiations of the equality of opportunity (EOP) principle. Similarly, [10] show that they can be viewed as special cases of a more general principle of fairness they call fair equality of chances (FEC). This way, they provide a framework through which the existing fairness criteria can be viewed. However, the conditions under which the existing fairness criteria map to EOP (or to FEC, respectively) are not always given. We cannot expect every application to fall neatly into one of these conditions and thus cannot expect to find a fitting fairness criterion among the ones already proposed in the group fairness literature.

These more general notions of fairness might be suitable to grasp the different existing notions of group fairness. However, they do not adequately represent the complexity of the distributive justice literature Kuppler et al. [33]. In this paper, we want to bridge the gap between fair machine learning and philosophical theories of distributive justice.

## 3   A framework for fairness evaluations based on distributive justice

As discussed in Section 2, current group fairness criteria have some serious shortcomings. Clearly, they do not reflect the full breadth of the literature on distributive justice [33]. To address this issue (at least partially), we propose a utility-based extension of group fairness. This section introduces this approach from a rather technical perspective. More details on its links to the literature on distributive justice can be found in [3]. Our approach is based on the observation that each decision system creates a *distribution* of utility among individuals and groups. Theories of distributive justice are concerned with the question of when such a distribution can be considered just. As we will later show, some of these theories can be mapped to classical group fairness concepts from the fair ML literature (see Section 4).

We consider a decision making system that takes binary decisions $D$ on decision subjects $DS$ of a given population $P$, based on a decision rule $r$. The decision rule assigns each individual $i \in P$ a binary decision $d_i \in \{0, 1\}$, applying the decision rule to some input data, which includes an unknown but decision-relevant binary random variable $Y$. It does not matter how the decision rule functions. It could, for example, be an automated rule that takes decisions based on predictions of $Y$ from an ML model or the decisions could be made by humans. We further assume that at least two social groups are defined, denoted with different values for the sensitive attribute $A$.

## 3.1 Utility of the decision subjects

As previously discussed, current definitions of group fairness only consider the decisions themselves, but not their consequences — even though the same decision could be beneficial for some and harmful for others [54]. Our approach explicitly considers the consequences of decisions, i.e., the resulting *utility* (or *welfare*), which could be positive in the case of a benefit or negative in the case of a harm. We model the consequences with a utility function $u$ which, in our binary context, may depend on both the decision $d_i$ and the value $y_i$ of $Y$.

The utility $u_{DS,i}$ of a decision subject $i$ is given by:

$$u_{DS,i} = w_{11} \cdot d_i \cdot y_i + w_{10} \cdot d_i \cdot (1 - y_i) + w_{01} \cdot (1 - d_i) \cdot y_i + w_{00} \cdot (1 - d_i) \cdot (1 - y_i), \quad (1)$$

where the utility weights $w_{dy}$ denote the four different utility values that might be realized for the four combinations of the random variables $Y$ and $D$.[1]

The utility $u_{DS,i}$ is a realization of a random variable $U_{DS}$. For assessing the fairness of a decision rule, we are interested in *systematic* differences between groups. Our framework is based on the assumption that such differences correspond to differentThis means that we are interested in the expectation values $E(U_{DS})$ of the individual utility, for different groups in $A$. Note that this is a normative choice and that other ways of comparing groups are imaginable, e.g., comparing their aggregated utilities.

## 3.2 Relevant groups to compare

Theories of distributive justice are typically concerned with individuals [48] while group fairness is concerned with socially salient groups. Group fairness focuses on comparisons of different groups as this is what theories of discrimination are concerned with [1]. This poses the question of how the comparison of individuals in distributive justice and the comparison of socially salient groups in group fairness can be combined? John Rawls's concept of "relevant positions" [42, §16, pp. 81-86] is a concept that unites both ideas. We view "relevant positions" as the groups whose expected utility we want to compare and refer to them as the relevant groups (to compare).[2] As defined in [3], relevant groups to compare have comparable moral claims[3] to receive the same utility, but probably do not receive the same utility. Our approach thus views the theories of distributive justice, which we introduced in Section 2, from the perspective of relevant groups to compare.

To be more specific, relevant groups are defined by two concepts: (1) *claims differentiator $J$*: What makes it the case that some people have the same claims to utility while others have different claims to utility?; (2) *causes of inequality* (resulting in socially salient groups $A$): What are the most likely causes of inequalities?

As described in [3], the claims differentiator identifies people who have equal moral claims. In other words, the utility should be distributed equally between these people. This means we only consider people with equal claims for our fairness evaluation.[4] Within the group of all individuals that have equal claims to utility (i.e., that are equal in their value for $J$) we specify groups that are unlikely to end up receiving equal utility, on average, based on the known causes of inequality (i.e., that are different in their value for $A$, which is sometimes also referred to as *protected attribute*). $J$ and $A$ define the relevant groups that group fairness criteria compare. For simplicity, we will assume that there are only two groups $A = \{0, 1\}$ that are unlikely to receive the same utility. It is, for example, common to expect individuals of a different race or gender to not derive the same utility from decision systems.

---

[1]In practice, however, one could use a much more specific utility function, using other attributes as well. A rather simple extension would also take $A$ into account and define these four utility weights for each group separately. That should be supported by an analysis of the inequality generated in the transition between the decision space and the utility space between different (socially salient or other) groups. In philosophy and economics, the work of Amartya Sen explains why resources do not always convert into the same capabilities (options to be and do) [49, pp. 21-23].

[2]This builds on Anonymous [4] which refers to relevant positions as "representative individuals".

[3]For a philosophical analysis of comparable moral claims to a good, see [25].

[4]This concept is similar to the *justifier* described in [36, 10].

In the next step, we want to compare the utilities of the relevant groups. Specifically, we will compare the expectation value of utility over all decisions made for a given population under a given a decision rule. We denote this as the expected utility that takes the relevant groups into account, $E(U_{DS}|J = j, A = a)$, where $J$ denotes the claims differentiator and $j$ corresponds to a possible value of the variable $J$, and $a \in A$ denotes the different socially salient group to be compared with each other. In our framework, assessing fairness means comparing relevant groups with the same $j$, but different $a$, with respect to the distribution of utility.

## 3.3 Patterns for a just distribution of utility

The claims differentiator $J$ tells us which individuals have equal moral claims to the utility distributed by the decision process. However, in some cases, an equal distribution of utility among the relevant groups (defined by $J$ and $A$) may not be the primary concern for justice (see below). Our approach offers different choices, which we refer to as *patterns of justice*. For each of them, we will briefly explain their normative view of what constitutes justice. For each pattern, we formulate a *fairness constraint* and a *fairness metric*: A fairness constraint is a mathematical formalization of a pattern of justice, which can either be satisfied or not. A fairness metric $F$, on the other hand, can measure the degree to which this criterion is fulfilled. Note that we construct fairness metrics for a binary $A = \{0, 1\}$. Therefore, all patterns of justice that we present compare the expected utility of two relevant groups: $A = 0 \wedge J = j$ (i.e., $E(U_{DS}|J = j, A = 0)$) and $A = 1 \wedge J = j$ (i.e., $E(U_{DS}|J = j, A = 1)$). However, the patterns of justice that we introduce here (egalitarianism, maximin, prioritarianism, sufficientiarianism) can easily be translated to cases of more groups.

In the following, we introduce only a few patterns of justice (representing fairness principles for the allocation of goods) that are widely discussed in philosophical literature. However, our utility-based definition of group fairness should in no way be seen as limited to these patterns. Our approach can easily be extended to other patterns of justice and one may also implement their own pattern of justice. Our goal here is simply to highlight a few popular patterns of justice and how they can be embedded in our approach.

### 3.3.1 Egalitarianism

Egalitarianism – as the name suggests – demands equality [5]. Egalitarianism as a broad concept does not, however, specify *what* should be equalized. This is subject of the *equality of what* debate initiated by Sen [48]. One could, for example, aim to equalize the opportunities (equality of opportunity) or outcomes (equality of outcomes).

**Fairness criterion**   The egalitarian fairness criterion is satisfied if the expected utility is equal for the relevant groups:

$$E(U_{DS}|J = j, A = 0) = E(U_{DS}|J = j, A = 1) \tag{2}$$

**Fairness metric**   The degree to which egalitarianism is fulfilled is measured as the absolute difference between the two groups' expected utilities (lower values are better):[5]

$$F_{\text{egalitarianism}} = |E(U_{DS}|J = j, A = 0) - E(U_{DS}|J = j, A = 1)| \tag{3}$$

### 3.3.2 Maximin

Maximin describes the principle that among a set of possible distributions, the one that maximizes the expected utility of the relevant group that is worst-off should be chosen [35]. In contrast to egalitarianism, inequalities are thus tolerated if the worst-off group benefits from them. This has been defended by Rawls in the form of the "difference principle" [42, 43].

**Fairness criterion**   The maximin fairness criterion is satisfied if there is no other possible distribution that would lead to a greater expected utility of the worst-off relevant group, which we denote by $U_{DS}^{worst-off} = min_{a \in A}\Big(E(U_{DS}|J = j, A = a)\Big)$. It thus requires that the decision rule $r'$ (which

---

[5]Here, we consider the absolute difference in expected utilities. Alternatively, we could also consider the ratio of the two expected utilities.

represents the decision taken for each individual) results in a $U_{DS}^{worst-off}(r')$ that is greater or equal than the $U_{DS}^{worst-off}(r)$ for any other decision rule $r$ from the set of all possible decision rules $R$:

$$U_{DS}^{worst-off}(r') \geq max_{r \in R}\left(U_{DS}^{worst-off}(r)\right) \tag{4}$$

**Fairness metric**  The degree to which maximin is fulfilled is measured as the value of the lowest expected utility between all relevant groups (higher values are better):

$$F_{\text{maximin}} = min_{a \in A}\left(E(U_{DS}|J=j, A=a)\right) \tag{5}$$

### 3.3.3 Prioritarianism

Prioritarianism describes the principle that among a set of possible distributions, the one that maximizes the weighted sum of utilities across all people [26]. In contrast to egalitarianism, inequalities are thus tolerated if they increase this weighted sum of expected utilities. In this weighted sum, the expected utility of the worst-off relevant groups is given a higher weight (the maximin principle can be seen as the extreme version of this as an infinite weight is given to the worst-off relevant groups).

**Fairness criterion**  The prioritarian fairness criterion is satisfied if there is no other possible distribution that would lead to a greater overall expected utility, which is measured as a weighted aggregation of the relevant groups' expected utilities, where the expected utility of the worst-off relevant group is given a higher weight. It thus requires that the decision rule $r'$ results in a weighted utility $\tilde{U}_{DS}(r') = k \cdot U_{DS}^{worst-off}(r') + U_{DS}^{better-off}(r')$ that is greater or equal than the $\tilde{U}_{DS}(r)$ for any other decision rule $r$ from the set of all possible decision rules $R$:

$$\tilde{U}_{DS}(r') \geq max_{r \in R}\left(\tilde{U}_{DS}(r)\right), \tag{6}$$

where $\tilde{U}_{DS}$ denotes the sum of decision subject utilities for all groups with a weight $k > 1$ applied to the worst-off group.

**Fairness metric**  The degree to which prioritarianism is fulfilled is measured as an aggregate of the (weighted) expected utilities (higher values are better):

$$\begin{aligned}
F_{\text{prioritarianism}} = {} & k \cdot min\left(E(U_{DS}|J=j, A=0), E(U_{DS}|J=j, A=1)\right) \\
& + max\left(E(U_{DS}|J=j, A=0), E(U_{DS}|J=j, A=1)\right)
\end{aligned} \tag{7}$$

### 3.3.4 Sufficientarianism

Sufficientarianism [50] describes the principle that there is a minimum threshold of utility that should be reached by everyone in expectation. Inequalities between relevant groups above this minimum threshold are acceptable according to this principle. Inequalities are thus tolerated as long as all groups achieve a minimum level of utility in expectation.

**Fairness criterion**  The sufficientarian fairness criterion is satisfied if all groups' expected utilities are above a given threshold $t$:

$$\forall a \in A \ \ E(U_{DS}|J=j, A=a)(r') \geq t \tag{8}$$

**Fairness metric**  The degree to which sufficientarianism is fulfilled is measured as the number of groups whose expected utility is above the given threshold $t$ (higher values are better):

$$F_{\text{sufficientarianism}} = \sum_{a \in A} T_a, \text{ where } T_a = \begin{cases} 1, & \text{if } E(U_{DS}|J=j, A=a) \geq t \\ 0, & \text{otherwise} \end{cases}$$

## 3.4 Extension of group fairness

Based on the mathematical framework outlined in this section, we suggest an extension of the current understanding of group fairness as described in Section 2. Instead of seeing group fairness as demanding equality between socio-demographic groups with respect to some value, we instead propose the following definition:

**Definition 1** (Group fairness). *Group fairness* is the just distribution of utility among relevant groups.

What makes a distribution just depends on the pattern of justice. Thus, our extended understanding of group fairness does not necessarily require equal expected utilities across groups. Furthermore, our definition ensures that only relevant groups are being compared (in the most familiar case, these correspond to socio-demographic groups).

*Group fairness criteria*, in our sense, specify when group fairness is satisfied by a decision-making system. From this, it follows that there are more group fairness criteria than previously acknowledged. This extension of group fairness criteria alleviates some of the criticisms of currently popular group fairness criteria as we will show in Section 5.

# 4 Relation to existing group fairness criteria

Existing group fairness criteria are special cases of the utility-based extension we propose. In this section, we formally show under which conditions our approach maps to existing group fairness criteria (see Table 1 for a summary of the results). In particular, we look at well-known group fairness criteria: (conditional) statistical parity, equality of opportunity, false positive rate (FPR) parity, equalized odds, predictive parity, false omission rate (FOR) parity, and sufficiency. The mathematical definitions of these criteria can be found in Table 2 in Appendix A. Furthermore, we show how the utility-based group fairness metrics relate to existing ones. In this section, we only demonstrate when our utility-based approach results in one of three often discussed group fairness criteria: statistical parity, equality of opportunity, and predictive parity. We refer the interested reader to the Appendix B.2 where we provide a similar mapping for other existing group fairness criteria.

The findings we present in this section extend the ones of [23], [36], and [10]. While [23] consider the distribution of *undeserved* utility (what they call the *difference between an individual's actual and effort-based utility*), [36] and [10] use the decision subject utility $U_{DS}$ to derive a morally appropriate group fairness definition. This is similar to our approach presented in this paper; however, they only consider two options $U_{DS} = D$ and $U_{DS} = Y$, while our approach allows for arbitrary functions $f$ for the utility: $U_{DS} = f(D, Y)$.

Statistical parity (also called demographic parity or group fairness [18]) is defined as $P(D = 1|A = 0) = P(D = 1|A = 1)$. For specific decision subject utility weights $w_{dy}$ and without any claims differentiator $J$, the condition of our utility-based fairness criteria derived from our framework is equivalent to statistical parity:

**Proposition 2** (Statistical parity as utility-based fairness). If the utility weights of all possible outcomes (as described in Section 3.1) do not depend on the group membership ($w_{dy} \perp a$), and $w_{11} = w_{10} \neq w_{01} = w_{00}$, then the egalitarian pattern fairness condition with $J = \emptyset$ is equivalent to statistical parity.

The formal proof of Proposition 2 can be found in Appendix B.1.1.

We use $w_{1y}$[6] to denote the decision subject utility associated with a positive decision ($D = 1$) and $w_{0y}$ to denote the decision subject utility associated with a negative decision ($D = 0$). As we showed above, requiring statistical parity can be equivalent to requiring the fulfillment of a utility-based group fairness criterion. However, even if the two criteria are equivalent, this is not necessarily true if we compare the group fairness metrics that specify the degree to which these two criteria are fulfilled, i.e., if we compare the degree to which statistical parity is fulfilled with the degree to which a utility-based fairness metric is fulfilled:

---

[6]Recall that utility weights are denoted by $w_{dy}$, where both $d$ and $y$ can take the value 0 or 1. For simplicity, we use $w_{1y}$ as a placeholder for utility weights of all outcomes with a positive decision ($d = 1$) and for individuals of any type ($y \in \{0, 1\}$), i.e., $w_{10}$ or $w_{11}$.

**Corollary 3** (Partial fulfillment of statistical parity in terms of utility-based fairness). Suppose that the degree to which statistical parity is fulfilled is defined as the absolute difference in decision ratios across groups, i.e., $|P(D = 1|A = 0) - P(D = 1|A = 1)|$. If the utility weights of all possible outcomes do not depend on the group membership ($w_{dy} \perp a$), and $w_{11} = w_{10} \neq w_{01} = w_{00}$ (i.e., $w_{1y} \neq w_{0y}$), and $J = \emptyset$, then the degree to which egalitarianism is fulfilled is equivalent to the degree to which statistical parity is fulfilled, multiplied by $|w_{1y} - w_{0y}|$.

The formal proof of Corollary 3 can be found in Appendix B.1.2. Intuitively, $F_{\text{egalitarianism}}$, which is derived from the utility-based fairness approach and represents the degree to which egalitarianism is fulfilled, can be seen as the degree to which statistical parity is fulfilled, weighted by the absolute difference in utility for the decision received (decision subject utility for a positive versus a negative decision).

Equality of opportunity (also called TPR parity) is defined as $P(D = 1|Y = 1, A = 0) = P(D = 1|Y = 1, A = 1)$, i.e., it requires parity of true positive rates (TPR) across groups $a \in A$ [20].

**Proposition 4** (Equality of opportunity as utility-based fairness). If $w_{11}$ and $w_{01}$ do not depend on the group membership ($w_{d1} \perp a$), and $w_{11} \neq w_{01}$, then the egalitarian pattern fairness condition with $J = Y$ and $j = \{1\}$ is equivalent to equality of opportunity.

The formal proof of Proposition 4 can be found in Appendix B.1.3. Compared to statistical parity, equality of opportunity only requires equal acceptance rates across those subgroups of $A$ who are of type $Y = 1$. This corresponds to the claims differentiator $j = \{1\}$ for $J = Y$. Thus, we simply require the utility weights $w_{11}$ and $w_{01}$ to be unequal and independent of $a$ (which means that the utility weights $w_{11}$ and $w_{01}$ are constant across groups). As is the case for statistical parity, there are differences when looking at the degree to which the two notions of fairness are fulfilled (equality of opportunity and the utility-based fairness under the conditions specified in Proposition 4):

**Corollary 5** (Partial fulfillment of equality of opportunity in terms of utility-based fairness). Suppose that the degree to which equality of opportunity is fulfilled is defined as the absolute difference in decision ratios for individuals of type $Y = 1$ across groups, i.e., $|P(D = 1|Y = 1, A = 0) - P(D = 1|Y = 1, A = 1)|$. If $w_{11}$ and $w_{01}$ do not depend on the group membership ($w_{d1} \perp a$), $w_{11} \neq w_{01}$, $J = Y$, and $j = \{1\}$, then the degree to which egalitarianism is fulfilled is equivalent to the degree to which equality of opportunity is fulfilled, multiplied by $|(w_{11} - w_{01})|$.

The formal proof of Corollary 5 can be found in Appendix B.1.4.

Predictive parity (also called PPV parity [9] or outcome test [51]) is defined as $P(Y = 1|D = 1, A = 0) = P(Y = 1|D = 1, A = 1)$, i.e., it requires parity of positive predictive value (PPV) rates across groups $a \in A$.

**Proposition 6** (Predictive parity as utility-based fairness). If $w_{11}$ and $w_{10}$ do not depend on the group membership ($w_{1y} \perp a$), and $w_{11} \neq w_{10}$, then the egalitarian pattern fairness condition with $J = D$ and $j = \{1\}$ is equivalent to predictive parity.

The formal proof of Proposition 6 can be found in Appendix B.1.5. Compared to equality of opportunity, predictive parity requires an equal share of individuals to be of type $Y = 1$ among those subgroups of $A$ who receive the decision $D = 1$. This corresponds to the claims differentiator $j = \{1\}$ for $J = D$. Thus, we simply require the utility weights $w_{11}$ and $w_{10}$ to be unequal and independent of $a$. As is the case for the other group fairness criteria, there are differences regarding the degree to which the two notions of fairness are fulfilled (predictive parity and the utility-based fairness under the conditions specified in Proposition 6):

**Corollary 7** (Partial fulfillment of predictive parity in terms of utility-based fairness). Suppose that the degree to which predictive parity is fulfilled is defined as the absolute difference in the ratio of individuals that are of type $Y = 1$ among all those that are assigned the decision $D = 1$ across groups, i.e., $|P(Y = 1|D = 1, A = 0) - P(Y = 1|D = 1, A = 1)|$. If $w_{11}$ and $w_{10}$ do not depend on the group membership ($w_{1y} \perp a$), $w_{11} \neq w_{10}$, $J = D$, and $j = \{1\}$, then the degree to which egalitarianism is fulfilled is equivalent to the degree to which predictive parity is fulfilled, multiplied by $|w_{11} - w_{10}|$.

The formal proof of Corollary 7 can be found in Appendix B.1.6.

Considering Table 1, we see that existing group fairness criteria have a narrow understanding of utility and do not tolerate inequalities, which can ultimately be harmful to already marginalized groups as

Table 1: Mapping of existing group fairness metrics to our utility-based approach under Egalitarianism

| Conditions | | | Equivalent fairness criterion |
|---|---|---|---|
| $U_{DS}$ weights (for groups $a \in \{0, 1\}$) | $J$ | $j$ | |
| $w_{11} = w_{10} \neq w_{01} = w_{00} \ \wedge \ w_{dy} \perp a$ | $\emptyset$ | - | Statistical parity |
| $w_{11} = w_{10} \neq w_{01} = w_{00} \ \wedge \ w_{dy} \perp a$ | $L$ | $l$ | Conditional statistical parity |
| $w_{11} \neq w_{01} \ \wedge \ w_{d1} \perp a$ | $Y$ | $\{1\}$ | Equality of opportunity |
| $w_{10} \neq w_{00} \ \wedge \ w_{d0} \perp a$ | $Y$ | $\{0\}$ | False positive rate parity |
| $w_{11} \neq w_{01} \wedge w_{10} \neq w_{00} \ \wedge \ w_{dy} \perp a$ | $Y$ | $\{0, 1\}$ | Equalized odds |
| $w_{11} \neq w_{10} \ \wedge \ w_{1y} \perp a$ | $D$ | $\{1\}$ | Predictive parity |
| $w_{01} \neq w_{00} \ \wedge \ w_{0y} \perp a$ | $D$ | $\{0\}$ | False omission rate parity |
| $w_{11} \neq w_{10} \wedge w_{01} \neq w_{00} \ \wedge \ w_{dy} \perp a$ | $D$ | $\{0, 1\}$ | Sufficiency |

previous work has shown [27]. Moreover, existing group fairness criteria embed assumptions about who has equal or different moral claims to utility. If we were to, for example, demand equalized odds for credit lending (where $D$ is the bank's decision to either approve a loan ($D = 1$) or reject it ($D = 0$), and $Y$ is the loan applicant's ability to repay the loan ($Y = 1$) or not ($Y = 0$)), we would make the following assumptions: People who are different in their ability to repay their loans have different claims to utility. We must thus equalize the expected utilities between people who are able to repay their loans and we must also equalize the expected utilities between people who are not able to repay their loans. However, the assumptions listed in Table 1 may not be met for all decision making systems. Our utility-based extension is thus necessary to implement other views of justice.

# 5   Discussion

As we have seen, existing group fairness criteria are special cases of our utility-based approach. This approach addresses several of the limitations of existing group fairness criteria that we discussed in Section 2.

**The "leveling down objection"**   The "leveling down objection" is a prevalent anti-egalitarianism argument [41, 17] saying that less inequality is not desirable if this requires lowering the better-off group's welfare to match the one of the worse-off group. On this basis, choosing egalitarianism as the pattern of justice has been criticized in the algorithmic fairness literature (see, e.g., [36, 27, 54]). Our approach allows using other patterns of justice, such as maximin, prioritarianism, or sufficientarianism (see Section 3.3). Other patterns that can be formalized as mathematical formulas may also be used. One could, for example, combine several patterns into one and require equal expected utilities across groups as long as none of the groups is better off than it would be without any fairness requirement. This would represent a combination of egalitarianism and a group-specific baseline threshold (similar to sufficientarianism), making a "leveling down" of the better-off group impossible and adhering to the Pareto principle. Therefore, our approach links group fairness to a much larger part of the literature on distributive justice than current group fairness criteria.

**No consideration of consequences**   Existing group fairness criteria only consider the distribution of *either $D$ or $Y$*. This could be interpreted as analyzing the distribution of utility but assuming that utility is equivalent to *either $D$ or $Y$* instead of, for example, the combination of $D$ *and $Y$*. Existing group fairness criteria thus represent a very confining definition of utility. Our approach acknowledges that the utility of the decision subjects does not only depend on the decision itself but also on other attributes such as one's ability to repay a loan or one's socioeconomic status (see, e.g., [24, 54, 11]. This is represented through the utility function described in Section 3.1.

**Limited set of fairness definitions**   Previous attempts to guide stakeholders in choosing appropriate fairness criteria have taken on the form of explicit rules, such as in [45, 37, 44]. Such rules, however, presuppose a limited set of fairness definitions between which stakeholders can choose. Instead, we provide a method to construct ad-hoc fairness criteria that reflect the values decided on by the stakeholders by combining the definition of the utility function for decision subjects (Section 3.1), the relevant groups to compare (Section 3.2) and the pattern for a just distribution of utility (Section 3.3).

Many important questions remain and may be the subject of future research: What are relevant trade-offs when imposing utility-based group fairness criteria as requirements? Optimal decision rules for existing group fairness criteria have been derived by [20, 16, 9] – do they change for the fairness criteria defined by our approach? Further, while our approach creates a link between group fairness and different theories of justice, it does not cover theories of distributive justice that are structurally different from the ones we discussed, e.g., Nozick's entitlement theory [39]. It is unclear how such theories could be represented in formalized fairness criteria. Moreover, there is a risk that decision makers simply use our approach to bluewash their decision making system, which they may claim to be "fair" and "unbiased" after coming up with a fairness criterion that neatly fits their own goals. This is an issue with other fairness criteria as well. Therefore, it is important to make the process of defining fairness criteria accessible to the public, so that decision subjects can get involved and hold decision makers accountable. This raises the question: with utility functions being notoriously hard to define [49, 19], how could our approach be accessible enough for practical use? What may be needed is a process for eliciting values from stakeholders. One may object that this makes group fairness criteria similarly difficult to implement as individual fairness and counterfactual fairness. Our response to this is that existing group fairness criteria might *seem* easier to use, but they still embed values and assumptions about the context in which they are used. Our approach helps to make these assumptions explicit.

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

## A  Existing group fairness criteria

Here, we briefly introduce the most discussed group fairness criteria. Table 2 list the parity requirements associated with these criteria. *Statistical parity* demands that the share of positive decisions

is equal between socio-demographic groups (defined by the sensitive attribute $A = \{0, 1\}$) [18] – this is only required for a set of so-called legitimate attributes $l \in L$ for the criterion *conditional statistical parity* [16]. *Equality of opportunity*, similarly, demands equal shares of positive decisions between socio-demographic groups, but only for those whose target variable is positive ($Y = 1$) [20] – thus, it is sometimes also referred to as true positive rate (TPR) parity. *Equalized odds* – sometimes also called *separation* – requires both equality of opportunity and FPR parity (which is similar to equality of opportunity, however, it is limited to individuals of type $Y = 0$). In contrast, *predictive parity* demands equal shares of individuals of type $Y = 1$ across socio-demographic groups, but only for those who received a positive decision $D = 1$ – thus, it is sometimes also referred to as positive predictive value (PPV) parity. *Sufficiency* requires both PPV parity and false omission rate (FOR) parity (which is similar to PPV parity, however, it is limited to individuals who received a negative decision $D = 0$).

Table 2: Existing group fairness metrics

| Fairness criterion | Parity requirement |
|---|---|
| Statistical parity | $P(D = 1|A = 0) = P(D = 1|A = 1)$ |
| Conditional statistical parity | $P(D = 1|L = l, A = 0) = P(D = 1|L = l, A = 1)$ |
| Equality of opportunity | $P(D = 1|Y = 1, A = 0) = P(D = 1|Y = 1, A = 1)$ |
| False positive rate parity | $P(D = 1|Y = 0, A = 0) = P(D = 1|Y = 0, A = 1)$ |
| Equalized odds | $P(D = 1|Y = y, A = 0) = P(D = 1|Y = y, A = 1)$, for $y \in \{0, 1\}$ |
| Predictive parity | $P(Y = 1|D = 1, A = 0) = P(Y = 1|D = 1, A = 1)$ |
| False omission rate parity | $P(Y = 1|D = 0, A = 0) = P(Y = 1|D = 0, A = 1)$ |
| Sufficiency | $P(Y = 1|D = d, A = 0) = P(Y = 1|D = d, A = 1)$, for $d \in \{0, 1\}$ |

# B  Mapping existing group fairness criteria to our utility-based approach

## B.1  Omitted proofs

### B.1.1  Proof of Proposition 2

Recall that the utility-based fairness following the pattern of egalitarianism requires equal expected utilities between groups:

$$E(U_{DS}|J = j, A = 0) = E(U_{DS}|J = j, A = 1) \tag{B.9}$$

Since there is no claims differentiator (i.e., $J = \emptyset$), this can be simplified to:

$$E(U_{DS}|A = 0) = E(U_{DS}|A = 1) \tag{B.10}$$

For $w_{11} = w_{10}$ and $w_{01} = w_{00}$, the decision subject utility (see Equation 1) is:

$$u_{DS,i} = w_{0y} + (w_{1y} - w_{0y}) \cdot d_i, \tag{B.11}$$

where $w_{1y}$ denotes the decision subject utility associated with a positive decision ($D = 1$) and $w_{0y}$ denotes the decision subject utility associated with a negative decision ($D = 0$). Thus, the expected utility for individuals of group $a$ can be written as:

$$E(U_{DS}|A = a) = w_{0y} + (w_{1y} - w_{0y}) \cdot P(D = 1|A = a). \tag{B.12}$$

If the utility weights of all possible outcomes do not depend on the group membership ($w_{dy} \perp a$), and $w_{1y} \neq w_{0y}$[7], then the utility-based fairness following the pattern of egalitarianism (see Equation B.10) requires:

$$\begin{aligned} w_{0y} + (w_{1y} - w_{0y}) \cdot P(D = 1|A = 0) &= w_{0y} + (w_{1y} - w_{0y}) \cdot P(D = 1|A = 1) \\ \Leftrightarrow (w_{1y} - w_{0y}) \cdot P(D = 1|A = 0) &= (w_{1y} - w_{0y}) \cdot P(D = 1|A = 1) \\ \Leftrightarrow P(D = 1|A = 0) &= P(D = 1|A = 1), \end{aligned} \tag{B.13}$$

where the last line is identical to statistical parity.

---

[7]If $w_{1y} = w_{0y}$, then the utility-based fairness following the pattern of egalitarianism would always be satisfied and the equivalence to statistical parity would not hold.

### B.1.2 Proof of Corollary 3

Recall that the degree to which egalitarianism is fulfilled is defined as $F_{\text{egalitarianism}} = |E(U_{DS}|J = j, A = 0) - E(U_{DS}|J = j, A = 1)|$ (see Equation 3). If the utility weights of all possible outcomes do not depend on the group membership ($w_{dy} \perp a$), and $w_{11} = w_{10} \neq w_{01} = w_{00}$ (i.e., $w_{1y} \neq w_{0y}$), $J = \emptyset$, this can be written as (see Equations B.10 and B.12):

$$
\begin{aligned}
F_{\text{egalitarianism}} &= |\, (w_{0y} + (w_{1y} - w_{0y}) \cdot P(D = 1|A = 0)) \\
&\quad - (w_{0y} + (w_{1y} - w_{0y}) \cdot P(D = 1|A = 1)) \,| \\
&= |\, ((w_{1y} - w_{0y}) \cdot P(D = 1|A = 0)) - ((w_{1y} - w_{0y}) \cdot P(D = 1|A = 1)) \,| \\
&= |(w_{1y} - w_{0y}) \cdot (P(D = 1|A = 0) - P(D = 1|A = 1)) |
\end{aligned}
\tag{B.14}
$$

where the last line corresponds to a multiplication of $|w_{1y} - w_{0y}|$ with the degree to which statistical parity is fulfilled.

### B.1.3 Proof of Proposition 4

Recall that the utility-based fairness following the pattern of egalitarianism requires equal expected utilities between groups:

$$
E(U_{DS}|J = j, A = 0) = E(U_{DS}|J = j, A = 1)
\tag{B.15}
$$

Since the claims differentiator is the same as the attribute $Y = 1$, i.e., $J = Y$ and the only morally relevant value of $Y$ is 1 (i.e., $j = \{1\}$), this can be simplified to:

$$
E(U_{DS}|Y = 1, A = 0) = E(U_{DS}|Y = 1, A = 1)
\tag{B.16}
$$

For $y_i = 1$, the decision subject utility (see Equation 1) is:

$$
u_{DS,i} = w_{01} + (w_{11} - w_{01}) \cdot d_i.
\tag{B.17}
$$

Thus, the expected utility for individuals of type $Y = 1$ in group $a$ can be written as:

$$
E(U_{DS}|Y = 1, A = a) = w_{01} + (w_{11} - w_{01}) \cdot P(D = 1|Y = 1, A = a).
\tag{B.18}
$$

If $w_{11}$ and $w_{01}$ do not depend on the group membership ($w_{d1} \perp a$), and $w_{11} \neq w_{01}$[8], then the utility-based fairness following the pattern of egalitarianism (see Equation B.16) requires:

$$
\begin{aligned}
w_{01} + (w_{11} - w_{01}) \cdot P(D = 1|Y = 1, A = 0) &= w_{01} + (w_{11} - w_{01}) \cdot P(D = 1|Y = 1, A = 1) \\
\Leftrightarrow (w_{11} - w_{01}) \cdot P(D = 1|Y = 1, A = 0) &= (w_{11} - w_{01}) \cdot P(D = 1|Y = 1, A = 1) \\
\Leftrightarrow P(D = 1|Y = 1, A = 0) &= P(D = 1|Y = 1, A = 1),
\end{aligned}
\tag{B.19}
$$

where the last line is identical to equality of opportunity.

### B.1.4 Proof of Corollary 5

Recall that the degree to which egalitarianism is fulfilled is defined as $F_{\text{egalitarianism}} = |E(U_{DS}|J = j, A = 0) - E(U_{DS}|J = j, A = 1)|$ (see Equation 3). If $w_{11}$ and $w_{01}$ do not depend on the group membership ($w_{d1} \perp a$), $w_{11} \neq w_{01}$, $J = Y$, and $j = \{1\}$, this can be written as (see Equations B.16 and B.18):

$$
\begin{aligned}
F_{\text{egalitarianism}} &= |\, (w_{01} + (w_{11} - w_{01}) \cdot P(D = 1|Y = 1, A = 0)) \\
&\quad - (w_{01} + (w_{11} - w_{01}) \cdot P(D = 1|Y = 1, A = 1)) \,| \\
&= |\, ((w_{11} - w_{01}) \cdot P(D = 1|Y = 1, A = 0)) \\
&\quad - ((w_{11} - w_{01}) \cdot P(D = 1|Y = 1, A = 1)) \,| \\
&= |(w_{11} - w_{01}) \cdot (P(D = 1|Y = 1, A = 0) - P(D = 1|Y = 1, A = 1)) |
\end{aligned}
\tag{B.20}
$$

where the last line corresponds to a multiplication of $|w_{11} - w_{01}|$ with the degree to which equality of opportunity is fulfilled.

---

[8]If $w_{11} = w_{01}$, then the utility-based fairness following the pattern of egalitarianism would always be satisfied and the equivalence to equality of opportunity would not hold.

 ### B.1.5  Proof of Proposition 6

Recall that the utility-based fairness following the pattern of egalitarianism requires equal expected utilities between groups:

$$E(U_{DS}|J = j, A = 0) = E(U_{DS}|J = j, A = 1) \tag{B.21}$$

Since the claims differentiator is the same as the decision $D = 1$, i.e., $J = D$ and the only morally relevant value of $D$ is 1 (i.e., $j = \{1\}$), this can be simplified to:

$$E(U_{DS}|D = 1, A = 0) = E(U_{DS}|D = 1, A = 1) \tag{B.22}$$

For $d_i = 1$, the decision subject utility (see Equation 1) is:

$$u_{DS,i} = w_{10} + (w_{11} - w_{10}) \cdot y_i. \tag{B.23}$$

Thus, the expected utility for individuals in group $a$ that are assigned the decision $D = 1$ can be written as:

$$E(U_{DS}|D = 1, A = a) = w_{10} + (w_{11} - w_{10}) \cdot P(Y = 1|D = 1, A = a). \tag{B.24}$$

If $w_{11}$ and $w_{10}$ do not depend on the group membership ($w_{1y} \perp a$), and $w_{11} \neq w_{10}$[9], then the utility-based fairness following the pattern of egalitarianism (see Equation B.22) requires:

$$\begin{aligned} w_{10} + (w_{11} - w_{10}) \cdot P(Y = 1|D = 1, A = 0) &= w_{10} + (w_{11} - w_{10}) \cdot P(Y = 1|D = 1, A = 1) \\ \Leftrightarrow (w_{11} - w_{10}) \cdot P(Y = 1|D = 1, A = 0) &= (w_{11} - w_{10}) \cdot P(Y = 1|D = 1, A = 1) \\ \Leftrightarrow P(Y = 1|D = 1, A = 0) &= P(Y = 1|D = 1, A = 1), \end{aligned} \tag{B.25}$$

where the last line is identical to predictive parity.

### B.1.6  Proof of Corollary 7

Recall that the degree to which egalitarianism is fulfilled is defined as $F_{\text{egalitarianism}} = |E(U_{DS}|J = j, A = 0) - E(U_{DS}|J = j, A = 1)|$ (see Equation 3). If $w_{11}$ and $w_{10}$ do not depend on the group membership ($w_{1y} \perp a$), $w_{11} \neq w_{10}$, $J = D$, and $j = \{1\}$, this can be written as (see Equations B.22 and B.24):

$$\begin{aligned} F_{\text{egalitarianism}} &= |\, (w_{10} + (w_{11} - w_{10}) \cdot P(Y = 1|D = 1, A = 0)) \\ &\quad - (w_{10} + (w_{11} - w_{10}) \cdot P(Y = 1|D = 1, A = 1))\,| \\ &= |\, ((w_{11} - w_{10}) \cdot P(Y = 1|D = 1, A = 0)) \\ &\quad - ((w_{11} - w_{10}) \cdot P(Y = 1|D = 1, A = 1))\,| \\ &= |(w_{11} - w_{10}) \cdot (P(Y = 1|D = 1, A = 0) - P(Y = 1|D = 1, A = 1))\,| \end{aligned} \tag{B.26}$$

where the last line corresponds to a multiplication of $|w_{11} - w_{10}|$ with the degree to which predictive parity is fulfilled.

## B.2  Mapping to other group fairness criteria

In Section 4, we mapped our utility-based approach to the three group fairness criteria statistical parity, equality of opportunity, and predictive parity. Here, we additionally show under which conditions our utility-based approach is equivalent to other group fairness criteria: conditional statistical parity, false positive rate parity, equalized odds, false omission rate parity, and sufficiency.

### B.2.1  Conditional statistical parity

Conditional statistical parity is defined as $P(D = 1|L = l, A = 0) = P(D = 1|L = l, A = 1)$, where $L$ is what [16] refer to as the *legitimate* attributes. Thus, conditional statistical parity requires equality of acceptance rates across all subgroups in $A = 0$ and $A = 1$ who are equal in their value $l$ for $L$, where $L$ can be any (combination of) feature(s) besides $D$ and $A$.

---

[9]If $w_{11} = w_{10}$, then the utility-based fairness following the pattern of egalitarianism would always be satisfied and the equivalence to predictive parity would not hold.

**Proposition 8** (Conditional statistical parity as utility-based fairness)**.** If the utility weights of all possible outcomes do not depend on the group membership ($w_{dy} \perp a$), and $w_{11} = w_{10} \neq w_{01} = w_{00}$, then the egalitarian pattern fairness condition with $J = L$ is equivalent to conditional statistical parity.

The proof of Proposition 8 is similar to the one of Proposition 2.

Under these conditions, the degree to which $F_{\text{egalitarianism}}$ is fulfilled is equivalent to the degree to which conditional statistical parity is fulfilled, multiplied by $|w_{1y} - w_{0y}|$. This could easily be proved – similar to the proof of Corollary 3 but with the conditions of the utility-based fairness stated in Proposition 8.

### B.2.2 False positive rate (FPR) parity

FPR parity (also called predictive equality [16]) is defined as $P(D = 1|Y = 0, A = 0) = P(D = 1|Y = 0, A = 1)$, i.e., it requires parity of false positive rates (FPR) across groups $a \in A$.

**Proposition 9** (FPR parity as utility-based fairness)**.** If $w_{10}$ and $w_{00}$ do not depend on the group membership ($w_{d0} \perp a$), and $w_{10} \neq w_{00}$, then the egalitarian pattern fairness condition with $J = Y$ and $j = \{0\}$ is equivalent to FPR parity.

For $y_i = 0$, the decision subject utility (see Equation 1) is:

$$u_{DS,i} = w_{00} + (w_{10} - w_{00}) \cdot d_i. \tag{B.27}$$

Thus, the expected utility for individuals of type $Y = 0$ in group $a$ can be written as:

$$E(U_{DS}|Y = 0, A = a) = w_0 + (w_{10} - w_{00}) \cdot P(D = 1|Y = 0, A = a). \tag{B.28}$$

Hence, we simply require the utility weights $w_{10}$ and $w_{00}$ to be unequal and independent of $a$. Then, the proof of Proposition 9 is similar to the one of Proposition 4.

If $w_{10}$ and $w_{00}$ do not depend on the group membership ($w_{d0} \perp a$), and $w_{10} \neq w_{00}$, then the degree to which $F_{\text{egalitarianism}}$ is fulfilled is equivalent to the degree to which FPR parity is fulfilled, multiplied by $|w_{10} - w_{00}|$. This could easily be proved – similar to the proof of Corollary 5.

### B.2.3 Equalized odds

Equalized odds (sometimes also referred to as separation [7]) is defined as $P(D = 1|Y = y, A = 0) = P(D = 1|Y = y, A = 1)$, for $y \in \{0, 1\}$.

**Proposition 10** (Equalized odds as utility-based fairness)**.** If the utility weights of all possible outcomes do not depend on the group membership ($w_{dy} \perp a$), $w_{11} \neq w_{01}$, and $w_{10} \neq w_{00}$, then the egalitarian pattern fairness condition with $J = Y$ and $j = \{0, 1\}$ is equivalent to equalized odds.

The conditions under which the utility-based fairness criteria is equivalent is shown separately for equality of opportunity (see Proposition 4) and FPR parity (see Proposition 9). Since equalized odds requires equality of opportunity and FPR parity, the the conditions for both fairness criteria must be met (i.e., $w_{dy} \perp a$), $w_{11} \neq w_{01}$, $w_{10} \neq w_{00}$, $J = Y$, and $j = \{0, 1\}$), so that the utility-based fairness constraint is equivalent to equalized odds.

### B.2.4 False omission rate (FOR) parity

FOR parity is defined as $P(Y = 1|D = 0, A = 0) = P(Y = 1|D = 0, A = 1)$, i.e., it requires parity of false omission rates (FOR) across groups $a \in A$.

**Proposition 11** (FOR parity as utility-based fairness)**.** If $w_{01}$ and $w_{00}$ do not depend on the group membership ($w_{0y} \perp a$), and $w_{01} \neq w_{00}$, then the egalitarian pattern fairness condition with $J = D$, and $j = \{0\}$ is equivalent to FOR parity.

For $d_i = 0$, the decision subject utility (see Equation 1) is:

$$u_{DS,i} = w_{00} + (w_{01} - w_{00}) \cdot y_i. \tag{B.29}$$

Thus, the expected utility for individuals in group $a$ that are assigned the decision $D = 0$ can be written as:

$$E(U_{DS}|D = 0, A = a) = w_{00} + (w_{01} - w_{00}) \cdot P(Y = 1|D = 0, A = a). \tag{B.30}$$

717  Hence, we simply require the utility weights $w_{01}$ and $w_{00}$ to be unequal and independent of $a$. Then,
718  the proof of Proposition 11 is similar to the one of Proposition 6.

719  If $w_{01}$ and $w_{00}$ do not depend on the group membership ($w_{0y} \perp a$), and $w_{01} \neq w_{00}$, then the degree
720  to which $F_{\text{egalitarianism}}$ is fulfilled is equivalent to the degree to which FoR parity is fulfilled, multiplied
721  by $|w_{01} - w_{00}|$. This could easily be proved – similar to the proof of Corollary 7.

### B.2.5  Sufficiency

723  Sufficiency is defined as $P(Y = 1|D = d, A = 0) = P(Y = 1|D = d, A = 1)$, for $d \in \{0, 1\}$ [7].

724  **Proposition 12** (Sufficiency as utility-based fairness)**.**  If the utility weights of all possible outcomes
725  do not depend on the group membership ($w_{dy} \perp a$), $w_{11} \neq w_{10}$, and $w_{01} \neq w_{00}$, then the egalitarian
726  pattern fairness condition with $J = D$ and $j = \{0, 1\}$ is equivalent to sufficiency.

727  The conditions under which the utility-based fairness criteria is equivalent is shown separately for
728  predictive parity (see Proposition 6) and FOR parity (see Proposition 11). Since sufficiency requires
729  predictive parity and FOR parity, the the conditions for both fairness criteria must be met (i.e.,
730  $w_{dy} \perp a$), $w_{11} \neq w_{10}$, $w_{01} \neq w_{00}$, $J = D$, and $j = \{0, 1\}$), so that the utility-based fairness
731  constraint is equivalent to sufficiency.

