# OpenReview forum: "Distributive Justice as the Foundational Premise of Fair ML: Unification, Extension, and Interpretation of Group Fairness Metrics"
_NeurIPS.cc/2022/Conference — NeurIPS 2022 Submitted_

### Official Review · Reviewer_Votd · 2022-06-21

**Rating:** 4
**Confidence:** 4
**Soundness:** 3 good
**Presentation:** 3 good
**Contribution:** 2 fair

**Summary:**

The paper presents a general framework for understanding group fairness metrics that is connected to relevant literature in philosophy underpinning the relevant values and moral perspectives. Specifically, the paper proposes to interpret group fairness as the “just distribution of utility among relevant groups”, as opposed to simply “equality (of decisions) between socio-demographic groups”. This highlights the several components of the proposed approach:
- utility (as opposed to decisions)
- some notion of justice (as opposed to necessarily equality)
- comparing relevant groups (as opposed to simply demographic groups)

In slightly more detail: The starting point of the proposed approach is to focus on the utility of the individuals (here referred to as decision subjects) and to introduce the concept of a “claims differentiator” J, which identifies individuals with equal claims utility for the sake of the evaluation. Technically,  this is a binary variable where now fairness means comparing the expected utilities of individuals with the same J=j but different group membership A=a. There is then the question of how to compare these quantities - here comes the component of “pattern of justice”. The paper considers several such patterns studied in the philosophical literature and studies their implications in terms of the implied fairness metrics. For example, “egalitarianism” would seek to equalize the above-mentioned quantities, but other principles (Maximin, Prioritarianism) could require different things.


**Questions:**

Minor points:
- The notation U_{DS,i} seems overly complex, why not simply use u_i?
- I couldn’t follow some of the choices for the degree to which fairness is fulfilled. For example in Maximin section, the choice of (5) seemed a bit weird - in particular, wouldn’t it be non-zero even if fairness is met? I would expect something like the difference in the terms in Eq (4).
- Technically I think it would be helpful to clarify some of the mathematical notation. For example the notion of independence is used throughout for what I understand are realizations (e.g. w_dy independent of a in Table 1), and I’m not sure this is well-defined.


**Limitations:**

Yes.

**Strengths And Weaknesses:**

Strengths: The paper is generally clearly written, and bridging the gap between philosophical ideas and fair ML is an important objective.

Weaknesses: The main weakness of the paper in my opinion is that the main contributions of the paper are w.r.t the authors’ portrayal of the notion of group fairness (“equality (of decisions) between socio-demographic groups”), which I’m not sure is a actually fair w.r.t the *current* state of the literature. Specifically:

1. Confusion between group fairness as a metric and group fairness as an objective: Even with the narrow interpretation of equality of outcomes (or some other statistic that depends on the outcomes), an  important distinction in the literature is between group fairness as a metric and group fairness as an objective. I (and I believe many of my colleagues) view group notions of fairness as an important red flag, that if not satisfied, suggests one should look into the entire algorithmic pipeline for issues - be it the problem specification, the data collection, etc. In particular this means that the “solution” may be something like collecting better data, which does not require making any group’s welfare worse off.
2. A narrow interpretation of group fairness: There are many more notions of fairness studied in the literature that (i) utility based, (ii) that take preferences into account, or (iii) that speak to notions of fairness that do not necessarily involve parity (such as making sure no groups is “worse off” than what it would be had there been no other groups).  None of them were mentioned in the paper, and I found the perspective that the proposed unifying view is novel to be hindered by this.


Another weakness is that while the approach sets out to expose the normative and moral assumptions under-pinning group fairness, there is a bit of a “pick and choose” nature to this, as some assumptions which I think are not inherent (and encode some moral assumptions) are taken for granted and not discussed. For example, it is stated that since we are interested in systematic differences between groups, fairness only means we are “interested in the expectation value E(U_DS) of the individual utility”. Isn’t this also an assumption? Two groups may very well be systematically different even if their expectations are similar (but their variance, for example, is not). Does this not come up in the philosophical literature?

---

> ### Author Response · Authors · 2022-08-02
> **Answer to reviewer Votd**
>
> We thank the reviewer for the questions and comments and we address each concern below.
>
> 1. We agree with the reviewer’s view that group fairness notions are an important red flag that can be used to audit a prediction-based decision making process. Our framework differentiates between fairness criteria and fairness metrics (see Section 3.3). While these criteria are conceptually similar (and, as we show in our paper, in some cases identical) to parity constraints brought forward by the algorithmic fairness community, we specify fairness metrics as a way to measure the degree to which a specific criterion is satisfied and thus as a useful tool for auditing. The group fairness criteria our framework suggests can be used as a constraint during model training or to post-process the algorithmic predictions. However, the fairness metric can also be improved with other methods (e.g., simply by using a different ML algorithm). Thus, our framework is agnostic to any specific de-biasing techniques. In that sense, it is important to mention that incorporating de-biasing techniques at any point of the algorithmic pipeline can result in a group being worse off. This has, for example, been shown in experimental results in [1] where an SVM has been trained using fairness constraints. However, we agree that making any group worse off by the attempt to increase fairness is by no means guaranteed. Our point is that it cannot be ruled out. In particular, collecting better data does not rule this out as societal inequalities persist even in “better” data, so de-biasing techniques might still be necessary to achieve a certain fairness criterion.
>
> 2. Many group fairness metrics that are discussed in the academic literature on algorithmic fairness require the parity of some statistical measure across groups. The framework we propose unifies those notions of group fairness. As pointed out by the reviewer, other notions of fairness have been studied. While our framework is compatible with those that concern groups of individuals (e.g., preference-based fairness [2, 3]), it is not compatible with notions of fairness that are defined on the individual level (e.g., envy-freeness [4] or combinations of individuals and group fairness [5]). We now clarify this limitation of the suggested framework in Section 1 of the updated version of the paper. For a future version of the paper, however, we aim to understand how these other metrics relate to our framework and to explain this in the paper should it be accepted.
>
> Furthermore, we emphasize in Section 3.1 that our framework is based on the assumption that systematic differences between groups are represented in their expectation values. We agree that this is a normative choice and highlighted this in the updated version of the paper (Section 3.1). Most moral theories avoid the issue of within-group variance as they focus on comparing individuals. There is, however, a model for comparing the expectation of groups which was proposed by Rawls and which is the model we are following with our framework.
>
> Regarding some of the minor points:
> Equation (5): F_maximin measures the degree to which maximin is fulfilled. If fairness is fulfilled, F_maximin would be non-zero. The specific value of F_maximin depends on the chosen utility weights, however, a higher score corresponds to more fairness. This fairness metric could be normalized to make comparing cases with different utility weights easier.
> Notation: Table 1 states that certain utility weights are required to be independent of a. This means that those utility weights are constant across groups a. We specified this notion in Proposition 4.
>
>
> Thank you again for your valuable comments. We are looking forward to any further feedback you might have and are happy to answer your follow-up questions.
>
>
> [1] Hu, L., & Chen, Y. (2020). Fair classification and social welfare. Proceedings of the 2020 Conference on Fairness, Accountability, and Transparency, 535–545.
>
> [2] Zafar, M. B., Valera, I., Rodriguez, M., Gummadi, K., & Weller, A. (2017). From parity to preference-based notions of fairness in classification. Advances in Neural Information Processing Systems, 30.
>
> [3] Kim, M. P., Korolova, A., Rothblum, G. N., & Yona, G. (2019). Preference-informed fairness. arXiv preprint arXiv:1904.01793.
>
> [4] Balcan, M. F. F., Dick, T., Noothigattu, R., & Procaccia, A. D. (2019). Envy-free classification. Advances in Neural Information Processing Systems, 32.
>
> [5] Speicher, T., Heidari, H., Grgic-Hlaca, N., Gummadi, K. P., Singla, A., Weller, A., & Zafar, M. B. (2018, July). A unified approach to quantifying algorithmic unfairness: Measuring individual & group unfairness via inequality indices. In Proceedings of the 24th ACM SIGKDD international conference on knowledge discovery & data mining (pp. 2239-2248).

---

### Official Review · Reviewer_aeFZ · 2022-07-10

**Rating:** 6
**Confidence:** 3
**Soundness:** 4 excellent
**Presentation:** 3 good
**Contribution:** 3 good

**Summary:**

The authors propose a unifying framework for group fairness metrics which exposes different issues and justice patterns as choices in a mathematical equation. The key idea is to use explicitly the notion of utility (and expectation of it) in the definitions of fairness. By doing this they show that not only it is possible to represent all the most important fairness metrics in their formulation, but it is possible to construct new, more complex forms of fairness which address common criticisms of fairness criteria.

**Questions:**

Are there fairness formalisms and practical methods which are not covered by the unifying framework? Which are those? Why?

**Limitations:**

The authors should be more clear whether there are fairness formalisms which are not representable in their framework and why.

**Strengths And Weaknesses:**

Strengths:

The article aims to be formal, mathematical, and precise. Within this goal, it works well, and reads as easily as it would be expected in this context. The authors make a good job of pointing to other articles (included in the additional material) which have a deeper philosophical perspective and are, to some extent, easier for a less technical audience to read.

Weaknesses:

It would be really good to have the authors' view of what the drawbacks of their definition are. Often, unifying frameworks leave out some methods used in practice, and it would be great to have the authors' own view about what they have left behind. If nothing is left outside their framework, the authors' should clearly state that claim.

---

> ### Author Response · Authors · 2022-08-02
> **Answer to reviewer aeFZ**
>
> We appreciate the positive evaluation as well as the constructive comments by the reviewer.
>
> We agree that the limitations of the framework have to be clarified and thank the reviewer for this important feedback. Indeed, some philosophical theories of justice do not neatly fit into our three-step process. Nozick's entitlement theory [1], for example, is structurally different from our framework because Nozick believed that theories of justice should not be based on patterns [2]. This idea stands in contrast to many popular theories of justice for which patterns are a core principle. Patterns are also central to our framework. Theories such as Nozick’s that fundamentally disagree with an aspect of our framework can thus not be covered by our framework and it is still unclear if or how theories such as his could be represented in formalized fairness criteria. We have highlighted this in Section 5 of the updated version of the paper.
>
> [1] Nozick, R. (1974). Anarchy, state, and utopia (Vol. 5038). new york: Basic Books.
>
> [2] Duignan, B. (2022, January 19). Robert Nozick. Encyclopedia Britannica. https://www.britannica.com/biography/Robert-Nozick

---

> > ### Comment · Reviewer_aeFZ · 2022-08-09
> > **Please include what is not covered**
> >
> > Thanks for the answer. Please include what is not covered in the final version of the paper.

---

### Official Review · Reviewer_cmCu · 2022-07-12

**Rating:** 6
**Confidence:** 3
**Soundness:** 2 fair
**Presentation:** 3 good
**Contribution:** 2 fair

**Summary:**

This paper proposes a general framework for evaluating the fairness of a decision-making system, based on concepts from theories of distributive justice. The authors first point out the limitations and shortcomings of the current fairness metrics used in the literature, namely “the levelling down objection”, “non-consideration of consequences”, and a limited set of fairness definitions in the literature of fairness in AI. The framework they propose is based on utility values for individuals based on different combinations of decision D and the variable Y. They show that most popular fairness metrics present in the literature can be interpreted as special cases of their proposed frameworks.

**Questions:**

Have the authors consider provide experimental comparison of the proposed metrics and the existing metrics?

**Limitations:**

The paper lacks empirical evidence for their claims. While discussing downsides of current metrics, providing specific cases for which their claims might hold true would strengthen their claims.
Similarly, in addition to the theoretical and qualitative approach of their framework, providing empirical evidence for why and when their framework can be helpful in quantifying its effectiveness.


**Strengths And Weaknesses:**

Originality: This paper brings an interesting and fresh perspective to the discussion of fairness concept in AI field. I lack the understanding of ethics studies from the philosophical field that is discussed in this paper.  Therefore, I cannot comment on the originality of the ideas from that perspective. However, I believe that this paper would bring in an original enough perspective to the AI community working on fairness and ethical AI topics and would contribute to and enrich the discussions.

Quality: The submission is technically sound. The claims appear to be theoretically supported but lacks the empirical evidence.

Clarity: The paper is well written and organized.

Significance: This paper addresses an important issue, since most fairness metrics currently under consideration in the fairness literature suffer from levelling down objections.

---

> ### Author Response · Authors · 2022-08-02
> **Answer to reviewer cmCu**
>
> We appreciate the positive evaluation as well as the constructive comments by the reviewer.
>
> To answer the question about experimental evidence: Existing papers already provide some experimental evidence for the drawbacks of existing group fairness metrics. [1], for example, shows the leveling down objection in experiments. We agree that in future work, it would be interesting to empirically validate the advantages of the case-specific fairness metrics that result from our framework against standard ones. However, the objective of this paper is not to provide empirical comparisons, but rather to provide a general, unifying framework and thus a theoretical foundation for fairness metrics in the algorithmic fairness literature.
>
> [1] Hu, L., & Chen, Y. (2020). Fair classification and social welfare. Proceedings of the 2020 Conference on Fairness, Accountability, and Transparency, 535–545. https://doi.org/10.1145/3351095.3372857

---

### Meta-Review · Area_Chair_HVCJ · 2022-08-27

**Recommendation:** Reject
**Confidence:** Less certain

**Metareview:**

The paper grounds several fairness notions used in machine learning in principles of distributive justice. The stated motivation is to understand the normative choices behind each and to combat the shortcoming of some of these notions. The main concerns of the reviews were that this grounding is very limited in terms of its scope and there is little actionable insight that follows. Furthermore, many of the connections have already been acknowledged in the literature, e.g. [11, 29]. Philosophical underpinnings of the sciences are very important, as they can help advance both the questions we ask and the answers we offer. The effort of this paper is thus appreciated. However, as it falls somewhat short of advancing either the philosophy or the science, it may be of limited significance to the community. To garner better appreciation of their work, the authors are advised to elaborate on how their grounding could guide the field (e.g., How could one make algorithmic fairness choices in light of this perspective? Have there been instances where the wrong choice was made (algorithmically) relative to the stated intent (normatively)? Are there limitations to this perspective, perhaps in terms of assumptions that should be challenged? etc.)

**Award:**

No

---

### Decision · Program_Chairs · 2022-09-14

Reject